# N6-Methyladenosine Methylome Profiling of Muscle and Adipose Tissues Reveals Methylase–mRNA Metabolic Regulatory Networks in Fat Deposition of Rex Rabbits

**DOI:** 10.3390/biology11070944

**Published:** 2022-06-21

**Authors:** Gang Luo, Shuhui Wang, Yaotian Ai, Jiapeng Li, Zhanjun Ren

**Affiliations:** College of Animal Science and Technology, Northwest A&F University, Xianyang 712100, China; luogang66@nwafu.edu.cn (G.L.); wangshuhui252@163.com (S.W.); 18581899494@163.com (Y.A.); lijiapeng@nwafu.edu.cn (J.L.)

**Keywords:** Rex rabbits, m^6^A modification, methylation, metabolic regulatory, fat deposition

## Abstract

**Simple Summary:**

N6-methyladenosine is the most prevalent internal form of modification found in recent years and plays an important role in gene regulation, which can regulate many physiological processes, such as fat deposition, immunity, and reproduction. The intramuscular fat content is an important problem to be solved in the development of animal husbandry. In order to find a way to increase the intramuscular fat content of Rex rabbit muscles, we explored the methylation modification genes related to fat deposition in Rex rabbit muscle and adipose tissue. We found 5 differential methylases and 12 key genes for methylation modification related to fat deposition between muscle and adipose tissues samples. In addition, five differential methylases were found to regulate adipogenesis by affecting the expression of screened genes in different ways. These findings provided a theoretical basis for our future research on the function of methylation modification during the growth of fat deposits and provided a new way to increase intramuscular fat in Rex rabbits.

**Abstract:**

N6-methyladenosine (m^6^A) is the most prevalent internal form of modification in messenger RNA in higher eukaryotes and plays an important role in cancer, immunity, reproduction, development, and fat deposition. Intramuscular fat is the main factor used to measure the meat quality of an animal. The deposition of intramuscular fat and perirenal fat increases with age. However, there is no data on m^6^A modification of Rex rabbits and its potential biological roles in adipose deposition and muscle growth. Here, we performed two high-throughput sequencing methods, m^6^A-modified RNA immunoprecipitation sequence (MeRIP-seq) and RNA sequence (RNA-seq), to identify key genes with m^6^A modification on fat deposition in the muscle and adipose tissues of Rex rabbits. Then, qRT-PCR was used to identify the differently methylated genes related to fat deposition. Our findings showed that there were 12,876 and 10,973 m^6^A peaks in the rabbit muscle and adipose tissue transcriptomes, respectively. Stop codons, 3′-untranslated regions, and coding regions were found to be mainly enriched for m^6^A peaks. In addition, we found 5 differential methylases and 12 key genes of methylation modification related to fat deposition between muscle and adipose tissues samples. The expression levels of six random key genes were significantly higher in the fat than that in the muscle of Rex rabbits at different stages (*p* < 0.01). Finally, five differential methylases were found to regulate adipogenesis by affecting the expression of screened genes in different ways. These findings provided a theoretical basis for our future research on the function of m^6^A modification during the growth of fat deposits.

## 1. Introduction

A rabbit is a kind of livestock with fast growth, high reproductive performance, and short generation interval. As an economically important domestic animal, rabbits have fewer fat deposits compared with other mammals, such as swine, cattle, and sheep. In addition, a study found that the fat deposition pattern of perirenal fat and intramuscular fat is the same, which increased with age [1]. Fat deposition exists in both muscle and adipose tissue, but the similarities and differences in the specific regulation mechanisms of fat deposition are not clear. Therefore, the study of rabbit visceral fat development is of great significance to the development of animal husbandry. An increasing number of people have suffered from obesity-related metabolic diseases in recent decades due to excessive intake of high-fat diets. Obesity increases the likelihood of numerous chronic diseases, including type 2 diabetes, hypertension, cardiovascular disease, and cancer in humans [2,3]. Due to the naturally low fat deposition during rabbit growth, rabbits are an ideal model for studying visceral adipose development and have important clinical value. However, there have been few studies addressing the regulatory mechanisms involved in rabbit fat growth and metabolism. Thus, deepening our understanding of the molecular mechanism of fat deposition is of major economic and human health importance.

N6-methylation on adenosine (m^6^A) is one of the most advanced and popular research directions in the field of life science, which plays an important role in internal mRNA modification of eukaryotes [4,5]. Studies found that m^6^A is reversible [6]. In addition, m^6^A regulates0 transcriptome at the RNA level via reversible RNA methylation [7]. The reversibility of m^6^A is mainly regulated by writing protein, reader proteins, and eraser proteins. The binding proteins [8], demethylases [8], and the methyltransferase complex [9] play the roles of reading, erasing, and writing, respectively, in m^6^A modification. Much more recently, more and more biological roles of m^6^A have been found, such as stability, localization, mRNA splicing, translation, and translation efficiency [8]. Furthermore, RNA m^6^A plays an important role in murine stem cells [10,11]. In addition, m^6^A modification also plays a key role in biological processes such as cellular differentiation, lipid accumulation, and energy metabolism [4]. Recently, it has been proposed that m^6^A regulates adipogenesis through mediating mRNA splicing [12]. It has been demonstrated that the fat mass- and obesity-associated gene (*FTO*) is one of the m^6^A RNA demethylases and regulates adipogenesis through the modulation of mitotic clonal expansion [13]. Methyltransferase-like 3 (*METTL3*), a key RNA methyltransferase, has been demonstrated to regulate neurogenesis [14], spermatogenesis [15,16], early embryonic development [17], and stem cell pluripotency in mice [17,18]. However, mRNA m^6^A modification regulation and genetic mechanisms of *METTL14*, *YTHDC1*, *YTHDC2*, and *HNRNPA2B1* regulating fat deposition in Rex rabbits are far from clear.

In this study, we aimed to explore the regulatory mechanism of m^6^A modification on fat deposition in Rex rabbit muscle and perirenal adipose tissue. First, we identified m^6^A peaks and differential genes related to methylase and fat deposition by MeRIP-seq and RNA-seq in muscle and adipose tissues of Rex rabbits. Then, six random key genes were selected to validate by performing qRT-PCR in the muscle and adipose tissue of Rex rabbits at different stages. Finally, based on previous studies, five methylases were summarized and analyzed, which can regulate 12 genes related to fat deposition through different ways. These results enhance our understanding of molecular mechanisms associated with m^6^A modification and provide a basis for us to verify the regulation mechanism of fat deposition in muscle and fat tissue in adipocytes and rabbits.

## 2. Material and Methods

### 2.1. Animals and Tissue Collection

Perirenal adipose tissues and longissimus lumborum were collected from three 0-day-old Rex rabbits for MeRIP-seq. 35-day-old, 75-day-old, and 165-day-old female Rex rabbits (*n* = 3) were used for RT-qPCR, which were raised under standard conditions at the Northwest A&F University farm (Yangling, Shanxi, China). Rabbits were slaughtered with minimal pain.

### 2.2. RNA Extraction and Fragmentation

Total RNA was extracted using TRIzol reagent (Invitrogen, Carlsbad, CA, USA) following the manufacturer’s procedure. The concentration and quality of the RNA was evaluated using Nano Drop 2000 UV-Vis Spectrophotometer (Thermo Scientific, Waltham, MA, USA). The RNA integrity was assessed by Bioanalyzer 2100 (Agilent, CA, USA) with RIN number > 7.0, and confirmed by electrophoresis with denaturing agarose gel. Poly (A) RNA is purified from 50 μg total RNA using Dynabeads Oligo (dT)25-61005 (Thermo Fisher, CA, USA) using two rounds of purification. Then the poly(A) RNA was fragmented under 86 °C for 7 min.

### 2.3. M^6^A Immunoprecipitation and Library Construction

The cleaved RNA fragments were incubated for 2 h at 4 °C with m^6^A-specific antibody (No. 202003, Synaptic Systems, Germany) in IP (Immunoprecipitation) buffer (50 mM Tris-HCl, 750 mM NaCl and 0.5% Igepal CA-630). Then the IP RNA was reverse transcribed to create the cDNA. An A-base was then added to the blunt ends of each strand, preparing them for ligation to the indexed adapters. and size selection was performed with AMPureXP beads. After the heat-labile UDG enzyme (Baltimore, MD, USA) treatment of the U-labeled second-stranded DNA, the ligated products were amplified with PCR and then undertook final extension at 72 °C for 5 min. The average insert size for the final cDNA library was 300 ± 50 bp. At last, we performed the 2 × 150 bp paired-end sequencing (PE150) on an illumina Novaseq™ 6000 (LC-Bio Technology CO., Ltd., Hangzhou, China) following the vendor’s recommended protocol.

### 2.4. RNA Extraction and cDNA Synthesis

RNAiso Plus reagent (TaKaRa, Shiga, Japan) was used to extract total RNA following the manufacturer’s instructions. RNA quality and concentration refer to previous studies [19]. In addition, we used the Prime Script RT reagent Kit (Takara, Japan) to synthesize the first-strand cDNA of total RNA according to the manufacturer’s protocol.

### 2.5. Primer Design and Quantitative Real-Time PCR

PCR primers were designed with the Premier 6 software (http://www.greenxf.com/soft/190969.html, accessed on 5 June 2020) to amplify the entire coding DNA sequence (CDS) according to the reference mRNA sequence of rabbit *LPL* gene, *SNAP23* gene, *APMAP* gene, *ADCY4* gene, *PCK2* gene, *MAP4K3* gene, and *β-actin* gene in the GeneBank (GCF_000003625.3)*. β-actin* was used as an internal control to normalize the copy number of each gene by the 2^−ΔΔCt^ method [20] after obtaining the Ct values of each reference gene. The primers information is in Table 1.

### 2.6. KEGG and Gene Screening

Based on a large amount of literature, we screened genes related to fat deposition and KEGG. In addition, the genes verified by RT-qPCR were randomly selected.

### 2.7. Quality Control, Mapping and Statistical Analysis

The fastp software (https://github.com/OpenGene/fastp, accessed on 19 May 2022) was used to remove the reads that contained adaptor contamination, low quality bases, and undetermined bases with default parameter. Then sequence quality of IP and input samples were also verified using fastp. Reads were aligned to the reference genome of rabbits using Tophat (v2.0.14). For each gene, the reads count in each window was normalized by the median count of all windows of that gene. MEME [21] (http://meme-suite.org, accessed on 19 May 2022) and HOMER [22] (http://homer.ucsd.edu/homer/motif, accessed on 19 May 2022) were used for de novo and known motif finding followed by localization of the motif with respect to peak summit. Called peaks were annotated by intersection with gene architecture using R package [23] ChIPseeker (https://bioconductor.org/packages/ChIPseeker, accessed on 19 May 2022). In addition, the differentially expressed mRNAs were selected with log2 (fold change) > 1 or log2 (fold change) < −1 and *p* value < 0.05 by R package edgeR (https://bioconductor.org/packages/edgeR, accessed on 19 May 2022). Then StringTie (https://ccb.jhu.edu/software/stringtie, accessed on 19 May 2022) was used to perform the expression level for all mRNAs from input libraries by calculating FPKM. We used the balltown package of R language to analyze the difference of genes. The results of RT-qPCR were assessed using the GraphPad Prism software 5.0 (La Jolla, a seaside town in San Diego, USA) and presented as mean ± standard deviation (SD). Differences in the mean values between the 2 groups were determined for significance with a Student’s *t*-test. *p* < 0.05 and *p* < 0.01 were deemed to be significant and highly significant, respectively.

## 3. Results

### 3.1. Sequencing Statistics and Quality Control

In the process of MeRIP seq, the raw data were trimmed to remove the adaptor and date of low quality, and the clean reads were obtained. As shown in Table 2, the effective reads accounted for 89.94%, 90.48%, 90.77%, 91.02%, 90.69%, and 90.62% in the MeRIP-seq library, respectively. Raw data reads and valid data reads obtained from fat and muscle tissue samples are shown in Table 2. All proportions of effective reads were higher than 90.17%. In addition, all proportions of bases with mass values ≥ 20 were higher than 97.92% (sequencing error rate less than 0.01), all proportions of bases with quality values ≥ 30 were higher than 93.90% (sequencing error rate less than 0.001), and the percentage of GC in adipose tissue was lower than that in muscle tissue.

### 3.2. Mapping Reads to the Reference Genome

By mapping the reads data to the reference genome, as shown in Table 3, the mapping ratio of valid data in IP samples fat and muscle were more than88.288%. The proportion of the least-unique mapped reads 58.20% in the m^6^A-seq library. In the RNA-seq library, the mapping ratio of valid reads in input samples of fat and muscle were more than 88.71%. The proportion of unique mapped reads was not less than 62.54%. The proportion of multi-mapped reads is the mapping ratio of valid reads minus the proportion of unique mapped reads in Table 3. According to the region classification of reference genome, the proportion of sequencing sequences located in the exon region was the highest and the proportion of sequencing sequences located in the intergenic region was the lowest in Figure 1.

### 3.3. Transcriptome-Wide Detection and Distribution of m^6^A Modification in Rex Rabbits

We found m^6^A modified unique sequence RRACH in the sequencing results (Figure 2A). In addition, the distribution trend of m^6^A peaks in fat and muscle was similar, but there was significant difference in 5′ UTR; CDs, or only the end of CDs and the beginning of 3′ UTR were higher in fat than in muscle (Figure 2B). To determine the conservation and consequent functional importance of m^6^A, the expression of methylome from two rabbit tissues were compared. We identified 11,961 common peaks that were present from both tissues (Figure 2C).

### 3.4. KEGG Pathway Analysis in Muscle and Adipose Tissue

Based on previous studies, we found that 35 KEGG pathways (Figure 3) are related to fat metabolism and meat quality among the 330 KEGG pathways, including regulation of lipolysis in adipocyte, the MAPK signaling pathway, the PPAR signaling pathway, aldosterone synthesis and secretion, and other signal pathways.

### 3.5. Gene Screening and Overview of m^6^A-Modified Genes

Based on sequencing results, we found that 10 genes are hypo-methylation, and 2 genes are hyper-methylation. The distribution of 12 genes on a chromosome and the positions of the peaks in genes are shown in Table 3. Further exploration of these gene peaks revealed that 5 gene peaks covered only one exon among the 12 genes (Table 4). The distance between the peak and TSS, the size of the exon or UTR region that the peak spans or covers, and the initiation site relative to the first methylation initiation site are shown in Table 5. By further exploring the relationship between methylation regulation and gene regulation, we found that 9 genes were up regulated when methylation was down regulated, whereas 1 gene was down regulated when methylation was up regulated (Table 5). In addition, for one gene, methylation and gene expression were down regulated simultaneously, and for the remaining one, both were up regulated at the same time.

### 3.6. Overview of Differentially Expressed of Methylase Genes and Genes Related to Fat Deposition and Meat Quality in Muscle and Adipose Tissue Samples

We further explored the expression of methylase and found that expression levels of *METTL14*, *ZC3H13*, *YTHDC1*, and *HNRNPA2B1* in muscle tissue were significantly lower than those in adipose tissue (*p* < 0.01), and the expression level of *YTHDC2* was lower than that in adipose tissue (*p* < 0.05) (Figure 4A). Compared with the expression of muscle tissue, the results of FPKM values showed lower mRNA expression levels of the *TNMD* gene and *RCAN2* gene in adipose tissue (*p* < 0.01) (Figure 4B). However, the expression levels of the *LPL* gene, *APMAP* gene, *SNAP23* gene, *PCK2* gene, *MAP4K3* gene, *ADCY3* gene, *JMJD1C* gene, *PDCD4* gene, *AQP7* gene, and *RPGRIP1L* gene in adipose tissue were significantly higher than these in muscle tissue (*p* < 0.01) (Figure 4B).

### 3.7. Validation of Six Randomly Genes Related to Fat Deposition and Meat Quality by RT-qPCR

The expression levels of the *LPL* gene, *SNAP23* gene, *PCK2* gene, *ADCY3* gene, *APMAP* gene, and *MAP4K3* gene in adipose tissue were significantly higher than in the muscle tissue of Rex rabbits at 35 days of age, 75 days of age, and 165 days of age (*p* < 0.01) (Figure 5A–F), which was consistent with the results of RNA-seq (Figure 6).

### 3.8. Validation of Six Randomly Genes Related to Fat Deposition and Meat Quality by RT-qPCR

As shown in Figure 7, we found that *METTL14*, *YTHDC1*, *ZC3H13*, *YTHDC2*, and *HNRNPA2B1* regulated the expression of genes such as the *LPL* gene, *APMAP* gene, *SNAP23* gene, *PCK2* gene, *MAP4K3* gene, *ADCY3* gene, *JMJD1C* gene, *PDCD4* gene, *AQP7* gene, *RPGRIP1L* gene, *TNMD* gene, and *RCAN2* gene through a variety of pathways based on previous studies.

## 4. Discussion

M^6^A, as a new method of gene modification which has attracted more and more attention. However, we found no research on Rex rabbits. In this study, we found many methylation enzymes, methylation modification genes, and m^6^A peaks through using MeRIP-Seq. The distribution of the m^6^A peak in genes is similar to that in mice and humans [24,25]. However, the enrichment of the m^6^A site in plants is very different from that in Rex rabbits [26,27]. These results indicate that m^6^A modification is conserved only in mammals. In addition, an m^6^A-modified unique sequence, RRACH [28], was abundant in the sequencing results, indicating that there were a large number of m^6^A-modified sites.

In this study, methylase *METTL14*, *ZC3H13*, *YTHDC1*, *HNRNPA2B1*, and *YTHDC2* exist simultaneously in adipose tissue and muscle tissue, and the expression in adipose tissue is significantly higher than that in muscle tissue (*p* < 0.01). Previous studies have shown that *METTL14* [29] played an important role in fat deposition. In addition, *YTHDC1*, *ZC3H13*, *YTHDC2*, and *HNRNPA2B1* affected fat deposition through *PTEN* [30,31], *AKT* [32,33], *AKT*, and *STAT3* [34,35], respectively. However, the pathways of *METTL14*, *YTHDC1*, *ZC3H13*, *YTHDC2*, and *HNRNPA2B1* in regulating fat deposition are still unclear.

In order to further explore the specific regulatory mechanism of m^6^A modification in fat deposition, we analyzed the sequencing data by KEGG pathway to deduce potential functions of m^6^A-modified genes and found many of genes related to fat deposition and meat quality that we screened appeared in these signaling pathways. For example, *LPL* appeared in the cholesterol metabolism pathway; *PCK2* appeared in glycolysis/gluconeogenesis and adipocytokine signaling pathway, PI3K-Akt signaling pathway, pyruvate metabolism, AMPK signaling pathway, and FOXO signaling pathways; *MAP4K3* appeared in the MAPK signaling pathway; *AQP7* appeared in the regulation of lipolysis in the adipocyte and PPAR signaling pathways; and *ADCY3* appeared in the calcium signaling, regulation of lipolysis in adipocyte, thermogenesis, aldosterone synthesis and secretion, dilated cardiomyopathy (DCM), insulin secretion, chemokine signaling, thyroid hormone synthesis, platelet activation, gastric acid secretion, salivary secretion, and apelin signaling pathways. In addition, many key genes regulating fat deposition and meat quality appeared in other pathways, such as *PTEN*, *leptin*, and *IL-6*. First, *METTL14* can regulate 4 of the 12 key genes. *METTL14* promoted the binding of pri-miRNA-19a and pri-miRNA-375 to *DGCR8* and subsequent transformation into mature miRNA-19a and miRNA-375 [8]. MiRNA-19a played an important role in regulating the *PTEN/AKT/pAKT* pathway [36]. The *PTEN*-regulating miR-26a is amplified in high-grade glioma and miR-26a potently induced apoptosis and downregulated the expressions of *MAP4K3* [37,38]. miR-375 increased insulin secretion and insulin increased the activity of NM-IIA in the *SNAP23* complex by decreasing the level of *SEPTIN7* [39,40]. *ZC3H13* regulates *AKT* via inactivating Ras–ERK signaling [41,42]. Insulin can inhibit the expression of *PCK2* by activating the *AKT/FOXO1* signaling pathway, which is one of the main ways for insulin to inhibit hepatic gluconeogenesis [43]. *FOXO1* mediated leptin on food intake and the central leptin–melanocortin pathway played a pivotal role in the regulation of obesity by *ADCY3* [44,45]. Second, *YTHDC1* can regulate 7 of the 12 key genes. *YTHDC1* increased *AKT* phosphorylation by promoting *PTEN* mRNA degradation [30]. As mentioned before, *PTEN* can regulate the *MAP4K3* gene, *PCK2* gene, and *ADCY3* gene by different pathways. In addition, *YTHDC1* facilitated the biogenesis of mature miR-30d via m^6^A-mediated regulation [46]. miR-30d suppressed the *PI3K*/*AKT* pathway to inhibit cell biological progression [47], and the *PI3K* signaling pathway played an important role in regulating *AQP7* expression [48]. At the same times Resistin up-regulates *LPL* expression through the *PPARγ*-dependent *PI3K*/*AKT* signaling pathway, and *PPAR-γ* could modulate *APMAP* function [8]. Besides, Cut-like Homeobox 1 (CUX1) expression was decreased by *PI3K* inhibitors [49] and *CUX1* regulated expression of the *retinitis pigmentosa* GTPase regulator-interacting protein-1-like (RPGRIP1L) gene [50]. Third, *HNRNPA2B1* regulated 3 of the 12 key genes. *HNRNPA2B1* can promote the *STAT3* pathway [34] and leptin-activated human B cells to secrete *IL-6* via the *JAK2*/*STAT3* signaling pathway [51]. *IL-6* mediated the transcription of *JMJD1C* by regulating *OCT-4* gene expression [52,53] and reduced the expression of *TNMD* [54]. Finally, *YTHDC2* regulated 3 of the 12 key genes. *YTHDC2* is targeted for (HIF-1α) [55] 9 and *HIF-1α* could inhibit the expression of *PDCD4* by upregulating the expression of miR-21 [56,57]. In addition, *YTHDC2* could physically bind to insulin-like growth factor 1 receptor (IGF1R) messenger RNA and promote translation initiation of *IGF1R* mRNA [32]. *IGF-IR*/*PKM2* regulates miR-148a/152 expression [58], and *CIRC-TTC3* binds to miR-148a to regulate *RCAN2* [59]. These results indicated 4 methylases can regulate 12 key genes in different ways.

In order to further explore the effect of methylase on fat deposition through the key genes we screened, we summarized studies on the regulation of 12 key genes on fat deposition. *APMAP* is single transmembrane arylesterase which plays a cardinal role in adipogenesis [60]. Lipid droplet size is increased through fusion of primordial droplets, and *SNARE* proteins, including the *SNAP23*, are involved in this process [61]. *PCK2* is responsible for gluconeogenesis with lactic acid as a substrate [62]. The *ADCY3* gene is associated with obesity and lipid metabolism [63,64]. *LPL* plays an important role in the differentiation and maturation of animal adipocytes and in controlling the distribution of triglycerides in fat and muscle [65]. The percentage of 3T3-L1 preadipocytes differentiated into adipocytes was significantly reduced by interfering with *MAP4K3* expression in 3T3-L1 cells [66]. *JMJD1C* promotes lipogenesis in vivo to increase hepatic and plasma triglyceride levels [67]. CNS abnormalities caused by *RPGRIP1L* haploinsufficiency may cause obesity in humans [68]. HFD-fed *PDCD4*^−/−^ mice displayed relatively normal adipocyte morphology [69]. *TNMD* gene polymorphism is closely related to obesity and glucose metabolism [70]. *RCAN2* knockout can resist obesity induced by age and high-energy food, which preliminarily reveals the role of the *RCAN2* gene in the regulation process of fat deposition [71]. *AQP7* participates in adipogenesis and regulates the transport of triglycerides after hydration [72]. So, we speculated that 12 genes can affect fat deposition and meat quality.

## 5. Conclusions

In summary, we found 5 methylases and 12 genes associated with fat deposition and meat quality that were methylated. The expression levels of six random key genes were significantly higher in the fat than that in the muscle of Rex rabbits at different stages. Finally, we verified that 5 methylases regulated adipogenesis by 12 key gene in varies signaling pathways base on previous studies. The study provided a theoretical basis for our future research on the function of methylation modification during the growth of fat deposition and provided a new way to increase intramuscular fat in Rex rabbits.

## Figures and Tables

**Figure 1 biology-11-00944-f001:**
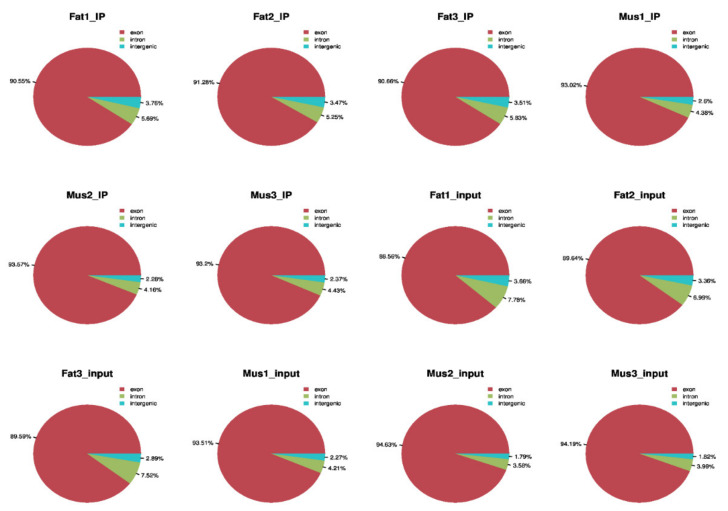
Refer to the genome to compare the regional distribution.

**Figure 2 biology-11-00944-f002:**
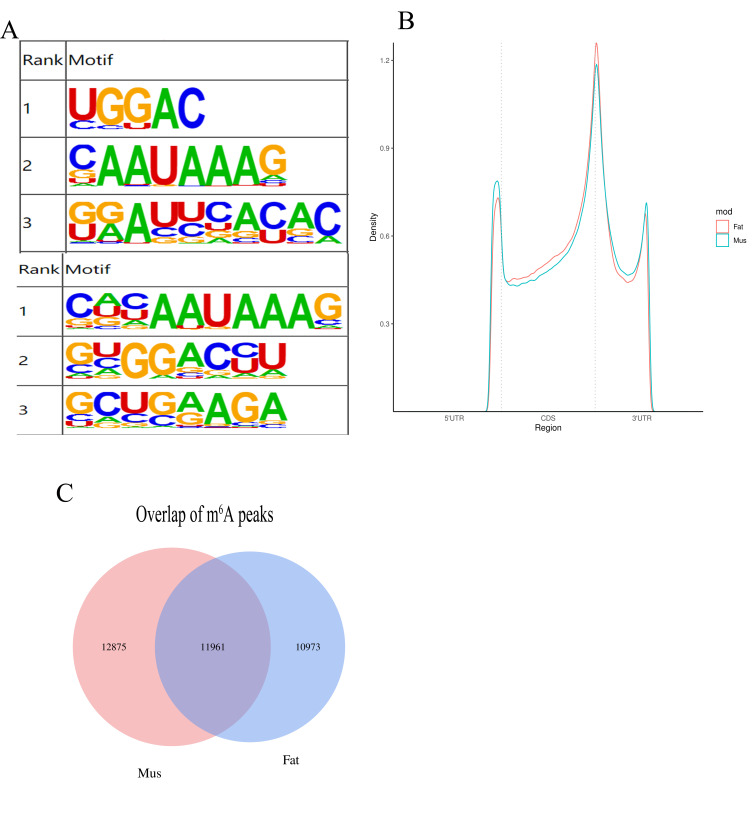
(**A**) Sequence logo showing the top motifs enriched across differential m^6^A peaks identified from muscle and fat samples; (**B**) distribution of m6A peaks across the length of mRNA; (**C**) overlap of m^6^A peaks from fat and muscle tissues. *p* ≤ 0.05 is statistically significant.

**Figure 3 biology-11-00944-f003:**
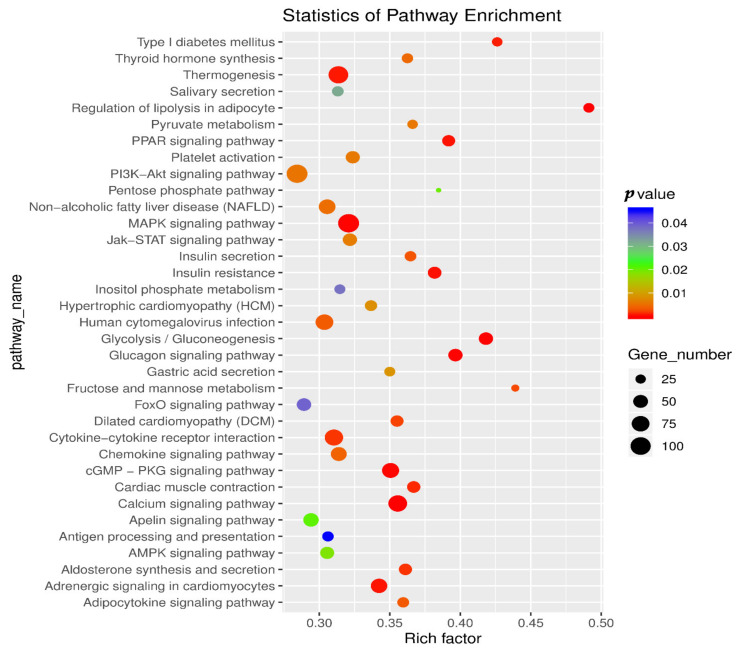
Enrichment pathway of m^6^A peak related to fat deposition.

**Figure 4 biology-11-00944-f004:**
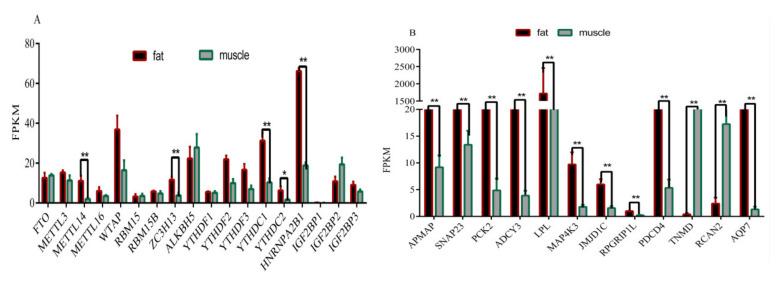
Overview of differentially expressed of methylase genes and key genes in muscle and adipose tissue samples. (**A**) FPKM of the methylase genes in muscle and adipose tissues; (**B**) FPKM of the key genes in muscle and adipose tissues (“*”, *p* ≤ 0.05; “**”, *p* ≤ 0.01).

**Figure 5 biology-11-00944-f005:**
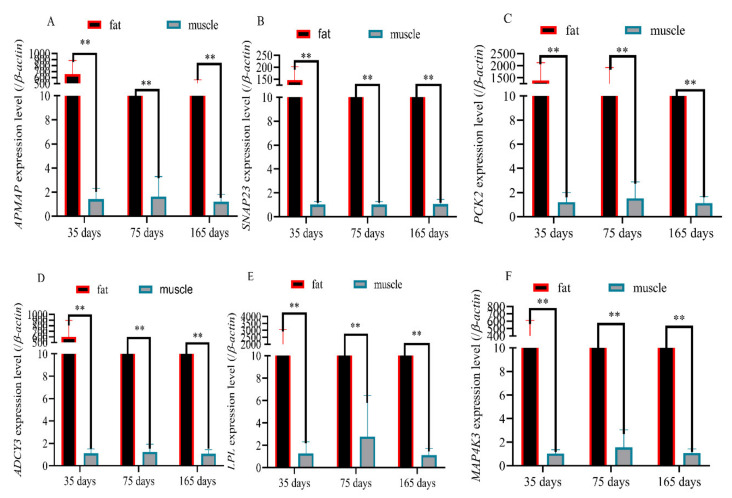
qPCR results of the 6 differentially m^6^A-modified genes in muscle and adipose tissue samples. (**A**) Expression levels of the *APMAP* gene in the muscle and adipose tissue of 35-, 75-, and 165-day-old Rex rabbits; (**B**) expression levels of the *SNAP23* gene in the muscle and adipose tissue of 35-, 75-, and 165-day-old Rex rabbits; (**C**) expression levels of the *PCK2* gene in the muscle and adipose tissue of 35-, 75-, and 165-day-old Rex rabbits; (**D**) expression levels of the *ADCY3* gene in the muscle and adipose tissue of 35-, 75-, and 165-day-old Rex rabbits; (**E**) expression levels of the *LPL* gene in the muscle and adipose tissue of 35-, 75-, and 165-day-old Rex rabbits; (**F**) expression levels of the *MAP4K3* gene in the muscle and adipose tissue of 35-, 75-, and 165-day-old Rex rabbits (“**”, *p* ≤ 0.01).

**Figure 6 biology-11-00944-f006:**
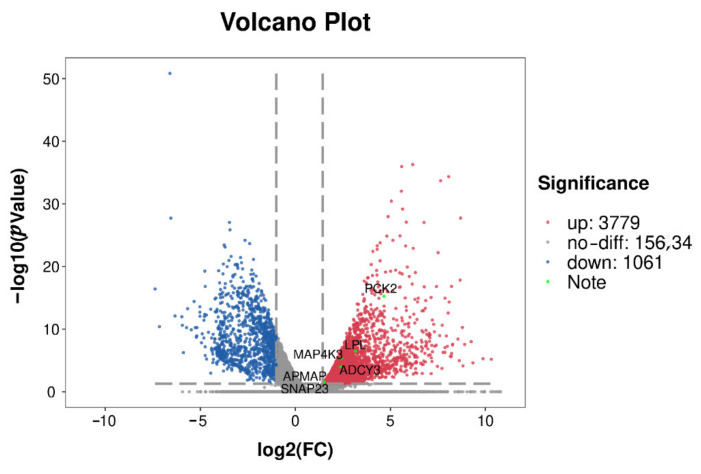
Volcanic map of differentially expressed genes.

**Figure 7 biology-11-00944-f007:**
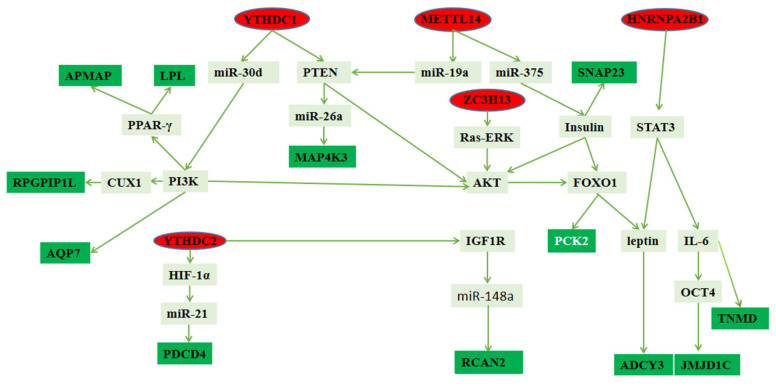
Pathway map of key genes regulated by methylase.

**Table 1 biology-11-00944-t001:** Primer pairs used for RT-qPCR.

Name	Primer Sequence	Temperature (°C)	Product Size (bp)	Gene ID
APMAP	5’-GCTGCTGGATTCTCCCATAG-3′	60	163	100339857
	5’-AAACATCACGTCCCCGATAT-3′			
*β* *-actin*	5’-GGAGATCGTGCGGGACAT-3′	61.4	318	100009272
	5’-GTTGAAGGTGGTCTCGTGGAT-3′			
SNAP23	5’-CCTGGCAATGTGGTGTCTAA-3′	59.5	250	100008776
	5’-TGGTGTCAGCCTTTTCTGTAAT-3′			
PCK2	5’-AACAGGAGGTGCGTGACATT-3′	60.2	250	100144327
	5’-GGGACAGGGAGTGTGAGAAG-3′			
ADCY3	5’-TGGGCGTCATGTCCTACTAC-3′	60	238	100343958
	5’-ACATTCTCGTGGCGGTACAT-3′			
LPL	5’-GACATTGGGGAGTTGCTGAT-3′	60.5	214	100340171
	5’ACTTGTCGTGGCATTTCACA-3′			
MAP4K3	5’-ATGTGGGGCACTCCAAACTA-3′	59.5	182	100356354
	5’-TGAAGTCTCGCCCTCTACTG-3′			

**Table 2 biology-11-00944-t002:** Summary of reads quality control.

Sample	Raw_Reads	Valid_Reads	Valid%	Q20%	Q30%	GC%
Fat1_IP	81562394	79590262	89.94	98.07	94.17	49.20
Fat2_IP	74268656	72697900	90.48	98.02	94.08	49.93
Fat3_IP	75635076	74194910	90.77	97.92	93.90	50.64
Mus1_IP	101998102	100309020	91.02	98.17	94.45	53.73
Mus2_IP	100858460	98910896	90.69	98.09	94.25	53.14
Mus3_IP	77539934	76213194	90.62	98.08	94.24	53.00
Fat1_input	67621338	66671322	90.59	98.09	94.16	48.59
Fat2_input	72342384	71424132	90.48	98.17	94.36	49.85
Fat3_input	72298544	71241220	90.17	98.16	94.40	51.05
Mus1_input	78891496	78053538	91.05	98.18	94.45	52.35
Mus2_input	102167202	100877896	90.49	98.04	94.18	53.68
Mus3_input	92062236	90789550	90.64	98.25	94.62	53.62

Parameter description: Q20%, proportion of bases with a mass value ≥ 20; Q30%, proportion of bases with a mass value ≥ 30; GC, proportion of GC content.

**Table 3 biology-11-00944-t003:** Summary of reads mapping to the rabbit reference genome.

Sample	Valid Reads	Mapped Reads	Unique Mapped Reads	Multi Mapped Reads
Fat1_IP	78626498	71401305 (90.81%)	50564672 (64.31%)	20836633 (26.50%)
Fat2_IP	71989454	65126746 (90.47%)	49953050 (69.39%)	15173696 (21.08%)
Fat3_IP	73616646	65572313 (89.07%)	49524600 (67.27%)	16047713 (21.80%)
Mus1_IP	100013482	83125913 (83.11%)	58211564 (58.20%)	24914349 (24.91%)
Mus2_IP	98559478	81835114 (83.03%)	59331136 (60.20%)	22503978 (22.83%)
Mus3_IP	75865960	62874564 (82.88%)	46707280 (61.57%)	16167284 (21.31%)
Fat1_input	65033698	60764764 (93.44%)	43533958 (66.94%)	17230806 (26.50%)
Fat2_input	69325074	64450485 (92.97%)	48831496 (70.44%)	15618989 (22.53%)
Fat3_input	70284120	64559775 (91.86%)	48476357 (68.97%)	16083418 (22.88%)
Mus1_input	77296586	69248163 (89.59%)	48632494 (62.92%)	20615669 (26.67%)
Mus2_input	99957966	88668893 (88.71%)	62516927 (62.54%)	26151966 (26.16%)
Mus3_input	65033698	60764764 (93.44%)	43533958 (66.94%)	17230806 (26.50%)

**Table 4 biology-11-00944-t004:** M^6^A peaks of 12 genes related to fat deposition.

Gene Name	log2(fc)	Methylation Regulation	Chromosome	Peak Region	Peak Star	Peak End	*p*-Value
APMAP	1.52	Hypo-methylation	65	3’ UTR	1,160,678	1,161,123	1 × 10^−^^42^
SNAP23	1.49	Hypo-methylation	17	Exon	29,652,649	29,656,542	5.01 × 10^−37^
PCK2	4.67	Hypo-methylation	17	3’ UTR	44,153,721	44,154,226	1.58 × 10^−33^
ADCY3	2.38	Hypo-methylation	2	Exon	173,934,224	173,934,819	5.01 × 10^−26^
LPL	3.16	Hypo-methylation	15	5’ UTR	4,554,062	4,554,301	0.0041
MAP4K3	2.43	Hypo-methylation	2	5’ UTR	146,895,441	146,925,862	0.008
JMJD1C	1.93	Hypo-methylation	18	Exon	23,014,029	23,014,960	0.00017
RPGRIP1L	2.18	Hypo-methylation	5	5’ UTR	10,022,614	10,027,508	0.0093
PDCD4	2.1	Hyper-methylation	18	5’ UTR	58,499,213	58,499,903	0.012
TNMD	−6.59	Hyper-methylation	22	Exon	88,793,707	88,793,990	0.05
RCAN2	−2.85	Hypo-methylation	12	Exon	35,531,621	35,531,680	0.027
AQP7	5.58	Hypo-methylation	1	3’ UTR	20,078,653	20,078,802	0.022

**Table 5 biology-11-00944-t005:** Candidate m^6^A-modified genes related to fat deposition and difference peaks.

Gene Name	Gene ID	M^6^A Regulation	Gene Regulation	Block Count	Block Sizes	Block Starts	Distance To TSS
APMAP	100339857	Down	up	1	446	0	41,647
SNAP23	100008776	Down	up	3	27, 145, 37	0, 2495, 3857	30,071
PCK2	100144341	Down	up	1	506	0	9040
ADCY3	100343958	Down	up	2	57, 94,	0, 502	72,693
LPL	100340171	Down	up	1	240	0	30
MAP4K3	100356354	Down	up	2	195, 15,	0, 30407	0
JMJD1C	100350438	Down	up	2	92, 298	0, 634	262,502
RPGRIP1L	100354520	Down	up	4	58, 92, 44, 47	0, 2264, 2782, 4848	239
PDCD4	100354557	up	up	2	54, 184	0, 507	0
TNMD	100125994	up	Down	2	72, 48	0, 236	0
RCAN2	100343905	Down	Down	1	60	0	303,927
AQP7	100350611	Down	up	1	150	0	11,448

## Data Availability

The datasets supporting the conclusions of this article are included within the article. The sequencing data has been uploaded to NCBI and the BioProject ID is PRJNA794064 (https://www.ncbi.nlm.nih.gov/Traces/study/?acc=PRJNA794064, accessed on 10 June 2022).

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
