# Peer review of "N6-Methyladenosine Methylome Profiling of Muscle and Adipose Tissues Reveals Methylase–mRNA Metabolic Regulatory Networks in Fat Deposition of Rex Rabbits"

_biology, 2022, doi:10.3390/biology11070944_

Round 1

Reviewer 1 Report

Abstract:

The abstract is well summarized. However, I suggest that the first few sentences be rewritten to make a clearer exposition about the relationship between m6A and intramuscular fat.

There are some minor spell mistakes:

Line 8: “m6A” – 6 should be in superindex, according to the rest of the manuscript

Line 11: change “increased” to “increase”

Line 17: “… tissue transcriptomes, respectively” add a comma

Introduction:

The introduction is well documented since there are a lot of references related to the goal of this study. However, there are some things to change:

Lines 28-29: please rewrite the sentence since it is not well understood.

Line 31: please add “a” before “study”

Line 32: I suggest changing “had the same law”; it seems colloquial

Line 34: change the comma to full-stop before “Therefore”

Line 47: use the acronym (m6A) since it has been already stated.

Line 48: change “In Addition” to “In addition”

Line 48: change “regulated” to “regulates”

Lines 54-55: use another link instead of “In addition” (i. e., moreover, furthermore, etc.)

Material and methods:

The material and methods used in this study are well explained and executed. Several questions arose:

-          Why the authors performed MeRIP-seq only to 0-day-old rabbits and no other ages?

-          For considering the integrity of RNA, what do the authors mean when stating that “RIN number >7.0” (Line 93)? A sample with RIN=7 has no a very good quality RNA.

There are some things to change or explain:

 Line 99: “m6A” – 6 should be in superindex, according to the rest of the manuscript.

Line 100: please specify what IP means.

Line 127: please specify the number of the references in GenBank

Line 132: please add the reference of 2-ΔΔCt method

Results

The results are well explained. There are some minor spell mistakes:

Line 163, 164, 178, 181: change “;” by full stop.

Line 164: “So” is colloquial; change to “Therefore” or similar.

Line 238: change “genes” to “gene”

Table 1: legend – please delete “in this study”. Explain what the acronyms mean (the same with the rest of the tables). Put capital letters when necessary (i.e. Name, Primer sequence)

Discussion

The discussion is very clear and well written. There are some minor spell mistakes:

Line 282: change “indicated” to “indicate”

Author Response

Thank you very much for your guidance to my manuscript. We have made relevant revisions according to the proposed guidance. I hope to be approved by editor and reviewer.

The specific reply is as follows:

Abstract:

The abstract is well summarized. However, I suggest that the first few sentences be rewritten to make a clearer exposition about the relationship between m6A and intramuscular fat.

We have corrected it

There are some minor spell mistakes:

Line 8: “m6A” – 6 should be in superindex, according to the rest of the manuscript

We have corrected it

Line 11: change “increased” to “increase”

We have corrected it

Line 17: “… tissue transcriptomes, respectively” add a comma

 We have corrected it

Introduction:

The introduction is well documented since there are a lot of references related to the goal of this study. However, there are some things to change:

Lines 28-29: please rewrite the sentence since it is not well understood.

We have corrected it

Line 31: please add “a” before “study”

 We have corrected it

Line 32: I suggest changing “had the same law”; it seems colloquial

We have corrected it

Line 34: change the comma to full-stop before “Therefore”

We have corrected it

Line 47: use the acronym (m6A) since it has been already stated.

We have corrected it

Line 48: change “In Addition” to “In addition”

We have corrected it

Line 48: change “regulated” to “regulates”

We have corrected it

Lines 54-55: use another link instead of “In addition” (i. e., moreover, furthermore, etc.)

 We have corrected it

Material and methods:

The material and methods used in this study are well explained and executed. Several questions arose:

   Why the authors performed MeRIP-seq only to 0-day-old rabbits and no other ages?

Because day 0 is a new stage of mammalian growth and development. The regulation and modification of 0-day-old rabbits are not affected by the environment. The influencing factors are relatively simple, which makes it easier to explore the truth of regulation.

In this study, m6A modified gene related to intramuscular fat deposition was screened through muscle and adipose tissue. The effect of age on fat deposition is expected to be studied later.

 For considering the integrity of RNA, what do the authors mean when stating that “RIN number >7.0” (Line 93)? A sample with RIN=7 has no a very good quality RNA.

Rin (RNA integrity number) value is an indicator of Agilent 2100 Bioanalyzer quality inspection. Generally speaking, rin>7 is RNA with high integrity

There are some things to change or explain:

 Line 99: “m6A” – 6 should be in superindex, according to the rest of the manuscript.

 We have corrected it

Line 100: please specify what IP means.

 We have corrected it

Line 127: please specify the number of the references in GenBank

We have corrected it

Line 132: please add the reference of 2-ΔΔCt method

We have corrected it

Results

The results are well explained. There are some minor spell mistakes:

Line 163, 164, 178, 181: change “;” by full stop.

 We have corrected it

Line 164: “So” is colloquial; change to “Therefore” or similar.

We have corrected it

Line 238: change “genes” to “gene”

 We have corrected it

Table 1: legend – please delete “in this study”. Explain what the acronyms mean (the same with the rest of the tables). Put capital letters when necessary (i.e. Name, Primer sequence)

  We have corrected it

Discussion

The discussion is very clear and well written. There are some minor spell mistakes:

Line 282: change “indicated” to “indicate”

We have corrected it

Reviewer 2 Report

                                                                      Manuscript Number: biology-1757860; Title: N6-methyladenosine methylome profiling of muscle and adi-2 pose tissues reveals methylase–mRNA metabolic regulatory 3 networks in fat deposition of Rex rabbits. Review:

M&M Needs more clarification, and there are a number of abbreviations in the body of the search that need explanation. Lines 77-78: Please paraphrase this part. I think you can write the numbers of animals used better than they are now.

Lines 80-81: I think it's best to combine the ethics part of experimental animals with ethical statement.

Table 1: please insert the accession NO.

Line 144: why you chose the Fisher test specifically.

Lines 144-157: Please separate the statistical procedure under a separate heading with more explanation.

Line 156: please delete this part “Two tail student’s t test was used to analyze the significance of the different levels” You can replace it with levels of statistical significance.

Lines 160-161: I think it is better to combine this part with the M&M part.

Lines 161-169: Please paraphrase this part and do not repeat writing the numbers in the table in the context.

Line 172: please correct the abbreviation of CG as it is GC in table 2

Table 2: please insert the key table to clarify the abbreviations.

Lines 176-182: Please paraphrase this part. Please don’t repeat the percentages in the context.

Tables 2&3: I think it's better to just show percentages with respect to valid reads in table 2 and both of Unique mapped reads and Multi mapped reads in table 3.

Figure 1A: Need more resolution.

Figure1B-C: Please review the statistical analysis again and verify the statistical significances.

Line 210: you said “Based on previous studies, we found 35 KEGG pathways” Did you collect data from previous studies and use the meta-analysis method or what? And if so, why didn't you mention it in the material?

Lines 210-222: Please delete this part. It is shown in figure 3, why you rewrite it again?

Figure 2: it is an interesting figure, but please add the level of significant 0.05 in figure key since it appears in dark blue color on the graphic.

Lines 209-210; 227-228; 260-261, these sentences must be shown in M&M not in results.

Lines 231, 235 and 238: tables number need revision.

Table 4: Was the p-value estimation intended to test the differences between Peak star and Peak end, and if so, why did you use the Fisher test? I think it is an incorrect test as it is used with categorical data when the sample size is small <5 as when comparing the number of hypo-methylation with hyper-methylation genes.

Results in table 6: I think that these data need to statistically analyze by an appropriate non parametric test to examine the significant differences between the numbers of up and down regulated genes in cases of methylation was up or down regulated.

Line 264: you said “which results were consistent with the results of 264 RNA-seq (Fig. 6).” You did not provide any explanation for this Fig., please provide a detailed explanation.

Figure 5: The normal distribution of the data must be tested in order to be able to choose the appropriate statistical method. The graph suggests a high dispersion of values.

The discussion part needs to be reformulated with an improvement in the presentation and discussion of the results, not repeating the information, and focusing on the important aspects of the research without over-narrating the information.

Line 310: Figure 7 and its related information must be moved to the results.

Also, the conclusion part needs to be reformulated.

Line 496: This reference must be converted into English   

.

Author Response

Thank you very much for your guidance to my manuscript. We have made relevant revisions according to the proposed guidance. I hope to be approved by editor and reviewer.

The specific reply is as follows:

Comments and Suggestions for Authors

 Manuscript Number: biology-1757860; Title: N6-methyladenosine methylome profiling of muscle and adi-2 pose tissues reveals methylase–mRNA metabolic regulatory 3 networks in fat deposition of Rex rabbits. Review:

M&M Needs more clarification, and there are a number of abbreviations in the body of the search that need explanation. Lines 77-78: Please paraphrase this part. I think you can write the numbers of animals used better than they are now.

We have corrected it

Lines 80-81: I think it's best to combine the ethics part of experimental animals with ethical statement.

We have corrected it

Table 1: please insert the accession NO.

We have corrected it

Line 144: why you chose the Fisher test specifically.

It is wrong. We have corrected it

Lines 144-157: Please separate the statistical procedure under a separate heading with more explanation.

The same statistical procedure has appeared many times in previous articles. The detailed explanation here will greatly increase the repetition rate of the article. The article where the detailed procedure is located has been added to this manuscript in the form of references

Line 156: please delete this part “Two tail student’s t test was used to analyze the significance of the different levels” You can replace it with levels of statistical significance.

We have corrected it

Lines 160-161: I think it is better to combine this part with the M&M part.

We have corrected it

Lines 161-169: Please paraphrase this part and do not repeat writing the numbers in the table in the context.

We have corrected it

Line 172: please correct the abbreviation of CG as it is GC in table 2

We have corrected it

Table 2: please insert the key table to clarify the abbreviations.

We have corrected it

Lines 176-182: Please paraphrase this part. Please don’t repeat the percentages in the context.

We have corrected it

Tables 2&3: I think it's better to just show percentages with respect to valid reads in table 2 and both of Unique mapped reads and Multi mapped reads in table 3.

We have reduced the space of unimportant figures, which are the basis of sample quality. In addition, the main body of this study is the sequencing results. I don't think it will have any impact on this manuscript if the tables are not merged

Figure 1A: Need more resolution.

We have corrected it

Figure1B-C: Please review the statistical analysis again and verify the statistical significances.

We have corrected it

Line 210: you said “Based on previous studies, we found 35 KEGG pathways” Did you collect data from previous studies and use the meta-analysis method or what? And if so, why didn't you mention it in the material?

We don't use the meta-analysis method. We artificially summarize the function of pathways by consulting a large number of studies.

Lines 210-222: Please delete this part. It is shown in figure 3, why you rewrite it again?

We have corrected it

Figure 2: it is an interesting figure, but please add the level of significant 0.05 in figure key since it appears in dark blue color on the graphic.

We have corrected it

Lines 209-210; 227-228; 260-261, these sentences must be shown in M&M not in results.

We have corrected it

Lines 231, 235 and 238: tables number need revision.

We have corrected it

Table 4: Was the p-value estimation intended to test the differences between Peak star and Peak end, and if so, why did you use the Fisher test? I think it is an incorrect test as it is used with categorical data when the sample size is small <5 as when comparing the number of hypo-methylation with hyper-methylation genes.

We have corrected it

Results in table 6: I think that these data need to statistically analyze by an appropriate non parametric test to examine the significant differences between the numbers of up and down regulated genes in cases of methylation was up or down regulated.

We further define the difference threshold according to the initial operation results. For example, consider the status parameter, select the OK option, and only consider comparing the number of reads and genes ≥ 10 in the comparison sample (note threshold 10)

Line 264: you said “which results were consistent with the results of 264 RNA-seq (Fig. 6).” You did not provide any explanation for this Fig., please provide a detailed explanation.

The results of RNA sequencing are shown in Figure 6, Consistent “The expression levels of LPL gene, SNAP23 gene, PCK2 gene, ADCY3 gene, APMAP gene and MAP4K3 gene in adipose tissue were significantly higher than that in muscle tissue of Rex rabbits” has been stated in the previous sentence

Figure 5: The normal distribution of the data must be tested in order to be able to choose the appropriate statistical method. The graph suggests a high dispersion of values.

We have corrected it

The discussion part needs to be reformulated with an improvement in the presentation and discussion of the results, not repeating the information, and focusing on the important aspects of the research without over-narrating the information.

We have corrected it

Line 310: Figure 7 and its related information must be moved to the results.

We have corrected it

Also, the conclusion part needs to be reformulated.

We have corrected it

Line 496: This reference must be converted into English   

We have corrected it

Round 2

Reviewer 2 Report

Lines 187-192: Please paraphrase this part and do not repeat writing the numbers in the table in the context. Lines 209-210: Please paraphrase this part and do not repeat writing the percentages in the table in the context Lines 238-239: please delete this sentence” To explore fat deposition of m6A modification, and KEGG pathway analysis were performed for differentially genes” and paraphrase Lines 256-257: please delete this sentence” To explore the role of m6A-modified genes on fat deposition and meat quality, we screened all differential genes according to the existing studies and found 12 differential genes related to fat deposition and meat quality” and paraphrase. Table 4: You must indicate the statistical analysis in the body of the research, or at least include in the table key the name of the test that you used in probability’s assessment. Lines 290-291: please delete this sentence” In order to verify the sequencing results, we randomly selected 6 key genes and verified them with RT-qPCR.” and paraphrase.

Figure 5: all statistical differences were at level 0.01, why did you mention the 0.05 level in the figure’s key? And, please mention the name of the test for measuring statistical differences

Author Response

Thank you very much for your guidance to my manuscript. We have made relevant revisions according to the proposed guidance. I hope to be approved by editor and reviewer.

The specific reply is as follows:

Lines 187-192: Please paraphrase this part and do not repeat writing the numbers in the table in the context. 

We have corrected it

Lines 209-210: Please paraphrase this part and do not repeat writing the percentages in the table in the context

We have corrected it

 Lines 238-239: please delete this sentence” To explore fat deposition of m6A modification, and KEGG pathway analysis were performed for differentially genes” and paraphrase

We have corrected it

Lines 256-257: please delete this sentence” To explore the role of m6A-modified genes on fat deposition and meat quality, we screened all differential genes according to the existing studies and found 12 differential genes related to fat deposition and meat quality” and paraphrase. 

We have corrected it

Table 4: You must indicate the statistical analysis in the body of the research, or at least include in the table key the name of the test that you used in probability’s assessment.

We have corrected it-

Lines 290-291: please delete this sentence” In order to verify the sequencing results, we randomly selected 6 key genes and verified them with RT-qPCR.” and paraphrase.

We have corrected it

Figure 5: all statistical differences were at level 0.01, why did you mention the 0.05 level in the figure’s key? And, please mention the name of the test for measuring statistical differences

We have corrected it
